# Pseudoclassical Dynamics of the Kicked Top

**DOI:** 10.3390/e24081092

**Published:** 2022-08-09

**Authors:** Zhixing Zou, Jiao Wang

**Affiliations:** 1Department of Physics and Key Laboratory of Low Dimensional Condensed Matter Physics (Department of Education of Fujian Province), Xiamen University, Xiamen 361005, China; 2Lanzhou Center for Theoretical Physics, Lanzhou University, Lanzhou 730000, China

**Keywords:** kicked top model, quantum resonance, pseudoclassical limit, dynamical entanglement

## Abstract

The kicked rotor and the kicked top are two paradigms of quantum chaos. The notions of quantum resonance and the pseudoclassical limit, developed in the study of the kicked rotor, have revealed an intriguing and unconventional aspect of classical–quantum correspondence. Here, we show that, by extending these notions to the kicked top, its rich dynamical behavior can be appreciated more thoroughly; of special interest is the entanglement entropy. In particular, the periodic synchronization between systems subject to different kicking strength can be conveniently understood and elaborated from the pseudoclassical perspective. The applicability of the suggested general pseudoclassical theory to the kicked rotor is also discussed.

## 1. Introduction

The study of quantum chaos, or quantum chaology [1], focuses on whether, how, and to what extent classical chaos may manifest itself in the quantum realm. In essence, it boils down to the general classical–quantum correspondence issue, as insightfully pointed out by Einstein at the very early development stage of quantum theory [2]. The quantum kicked rotor, presumably the best known paradigm of quantum chaos, was first introduced by Casati et al. in their seminal study that opened this field [3]. After four decades of investigation, the richness of this paradigmatic model appears to be surprising. Far beyond quantum chaos, it has also been realized that this model may play a unique role in other fundamental problems, such as Anderson localization (transition) [4,5,6,7] and the quantum Hall effect [8,9,10]. Centering around the kicked rotor, an expanded overlapping field encompassing all these relevant problems is emerging [11].

In contrast to its richness, another advantage of the kicked rotor lies in its simplicity, featuring only a single point particle on a circle subject to the stroboscopic external interaction, which makes the study of this model much more convenient than most others. An exception is the kicked top model [12], which has a finite Hilbert space, so that it is even more favorable for research. Interestingly and importantly, these two models usually demonstrate different aspects of quantum chaos in a complementary way. With all these advantages, they are often the first ideal candidates for probing new notions. In recent years, interesting notions having been intensively investigated range widely, from the out-of-time-order correlations [13,14,15], to the dynamical entanglement [15,16,17,18,19,20], the non-Hermitian properties [21,22], and so on.

The dynamical entanglement is devised to capture the decoherence process of a quantum system when being coupled to the environment. It has distinct characteristics if the system’s classical counterpart is chaotic. The kicked top has the spin algebra symmetry and, as such, it can be regarded as a composite of identical qubits. An additional advantage due to such a multiqubit interpretation is that, for studying the dynamical entanglement, there is no need to introduce the environment. It has been shown both theoretically and experimentally that, indeed, the dynamical entanglement may serve as a diagnosis of quantum chaos in the kicked top model [20,23].

However, as far as we know, in most previous studies of the kicked top, only a narrower range of comparatively weak kicking strength has been investigated, leaving its properties unexplored yet in a wider range of stronger kicking strength. The reason might be that, for the classical kicked top, the transition from regular to globally chaotic motion occurs at a rather weak kicking strength. When the system is already globally chaotic, further increasing the kicking strength would not result in any qualitatively new properties. Accordingly, due to quantum–classical correspondence, it is reasonable to conjecture that this would also be the case in the quantum kicked top in the semi-classical limit. Nevertheless, as illustrated in a recent study where measures of quantum correlations were studied [18], the quantum kicked tops at certain different kicking strengths may synchronize, in clear contrast to their classical counterparts.

In fact, it is worth noting that in the kicked rotor, the similar classical–quantum non-correspondence phenomenon, termed quantum resonance [24], has been recognized and studied ever since the beginning of the quantum chaos field. Later, it has also been realized that the properties of the system when being detuned from the quantum resonance condition can even be understood in a classical way through the so-called pseudoclassical limit [25,26], rather than the conventional semi-classical limit. This reminds us to consider whether the synchronization observed in the quantum kicked top may have any underlying connections to quantum resonance and the pseudoclassical limit. In this work, we will provide a positive answer to this question. In particular, we will suggest a more general scheme of the pseudoclassical limit that involves more information of the quantum dynamics, which allows us to successfully apply it not only to the kicked top, but also to the kicked rotor. When being applied to the kicked rotor, the previously developed pseudoclassical scheme is found to be a special case of the suggested one.

In the following, we will briefly describe the kicked top model in Section 2 first. Next, in Section 3, we will discuss the quantum resonance condition for the quantum kicked top and develop the pseudoclassical theory. The properties of the system adjacent to the quantum resonance condition will be discussed in detail with two illustrating cases in Section 4. In particular, the numerical studies and the comparison with the pseudoclassical theory will be presented. In Section 5, the properties of dynamical entanglement will be investigated from the perspective of the pseudoclassical limit. Finally, we will summarize our work and discuss its extension to the kicked rotor in Section 6.

## 2. The Kicked Top Model

The Hamiltonian of the kicked top model is [12]
$$H=\alpha J_{x}+\frac{\beta }{2j}J_{z}^{2}\sum_{n=-\infty }^{\infty }\delta \left(t-n\right),$$
where $J_{x}$, $J_{y}$, and $J_{z}$ are the angular momentum operators respecting the commutations $\left[J_{\lambda },J_{\mu }\right]=i\epsilon_{\lambda \mu \nu }J_{\nu }$ (the Planck constant *ℏ* is set to be unity throughout) and *j* is an integer or half-integer related to the dimension of the Hilbert space *N* as N=2j+1. The first term in *H* describes the procession around the *x* axis with angular frequency α, while the second term accounts for a periodic sequence of kicks with an overall kicking strength β (the period of kicking is set to be the unit of time). In the following, we will restrict ourselves to the case of integer *j*, but the discussions can be extended straightforwardly to that of half-integer *j*. Since the Hamiltonian is time-periodic, the evolution of the system for a unit time, or one step of evolution, can be fulfilled by applying the Floquet operator
(1)$$U=\exp\left(-i\frac{\beta }{2j}J_{z}^{2}\right)\exp\left(-i\alpha J_{x}\right)$$
to the present state. Obviously, *U* does not change under the transformation β→β+4jπ, implying that the properties of the quantum kicked top have a periodic dependence on the kicking strength β of period 4jπ. Thus, a better understanding of the quantum kicked top calls for investigations covering such a period.

In the semi-classical limit j→∞, following the Heisenberg equations, the one-step evolution of the system reduces to the following map [27]:(2)$$\begin{array}{cc} X^{\prime }= & X\cos[\beta (Y\sin\alpha +Z\cos\alpha )]-(Y\cos\alpha \\ \; & -Z\sin\alpha )\cdot \sin[\beta (Y\sin\alpha +Z\cos\alpha )],\\ Y^{\prime }= & X\sin[\beta (Y\sin\alpha +Z\cos\alpha )]+(Y\cos\alpha \\ \; & -Z\sin\alpha )\cdot \cos[\beta (Y\sin\alpha +Z\cos\alpha )],\\ Z^{\prime }= & Y\sin\alpha +Z\cos\alpha , \end{array}$$
with the normalized variables $X=J_{x}/j$, $Y=J_{y}/j$, and $Z=J_{z}/j$. This map defines the classical kicked top. Physically, this map describes the process of rotating the top along the *x* axis for an angle of α first to reach the intermediate state $\left(\tilde{X},\tilde{Y},\tilde{Z}\right)$, followed by further rotating it around $\tilde{Z}$ by $\beta \tilde{Z}$, which is the same as the quantum Floquet operator.

Note that the state (X,Y,Z) can be viewed as a point on the surface of a unit sphere. Therefore, it can be represented equivalently by two angles, denoted as Θ and Φ, via the coordinate transformation (X,Y,Z)=(sinΘcosΦ,sinΘsinΦ,cosΘ). For the sake of convenience, we denote map (Equation 2) in terms of Θ and Φ as
(3)$$\left(\Theta^{\prime },\Phi^{\prime }\right)=F\left(\Theta ,\Phi ;\alpha ,\beta \right),$$
where ($\Theta^{\prime }$, $\Phi^{\prime }$) is the state equivalent to $\left(X^{\prime },Y^{\prime },Z^{\prime }\right)$.

In order to make a close comparison between the quantum and the classical dynamics, we invoke the spin coherent state in the former, which has the minimum uncertainty in a spin system. A spin coherent state centered at (Θ,Φ), denoted as |Θ,Φ〉, can be generated from the angular momentum eigenstate |j,j〉 as
$$|\Theta ,\Phi \rangle =\exp\left(i\Theta [J_{x}\sin\Phi -J_{y}\cos\Phi ]\right)|j,j\rangle .$$

Here, |j,j〉 satisfies that $\left(J_{x}^{2}+J_{y}^{2}+J_{z}^{2}\right)|j,j\rangle =j\left(j+1\right)|j,j\rangle$ and $J_{z}|j,j\rangle =j|j,j\rangle$. The classical counterpart of |Θ,Φ〉 is the point (Θ,Φ) on the unit sphere.

## 3. The Pseudoclassical Theory

### 3.1. Quantum Resonance in the Kicked Top

The concept of quantum resonance was first introduced in the kicked rotor model. The Floquet operator for the kicked rotor is
$$U_{R}=\exp\left(-i\frac{p^{2}}{2}T\right)\exp\left(-iK\cos\theta \right),$$
where *T* and *K* are two parameters, θ is the angular displacement of the rotor, and *p* is the corresponding conjugate angular momentum. If T=4πr/s with *r* and *s* as two coprime integers, except for the cases of an odd *r* and s=2, the asymptotic growth in energy is quadratic in time, corresponding to a linear spreading of the wavepacket in the angular momentum space. This phenomenon is referred to as “quantum resonance” [24], since it is caused by the pure quantum effect, with no connections to the classical dynamics. Otherwise, the energy would undergo a linear growth stage, corresponding to the diffusive spread of the wavepacket in the angular momentum space, before it saturates due to quantum interference, which is known as the so-called dynamical localization [28]. For the special case of an odd *r* and s=2, it follows that $U_{R}^{2}=1$. Namely, the quantum dynamics is periodic of period two, which is referred to as “quantum antiresonance”.

There is an implicit connection between the kicked rotor and the kicked top. By assuming the scaling
T=β/jandK=αj,
it has been shown that the kicked rotor emerges as a limit case of the kicked top [27]. Given this, the notion of quantum resonance can be extended to the kicked top by assigning the quantum resonance condition that
$$\beta =4j\pi \frac{r}{s}$$
with coprime *r* and *s*. Indeed, under this condition, the kicked top has similar properties to the kicked rotor in quantum resonance. For example, for β=4jπ, the Floquet operator reduces to $U=\exp\left(-i\alpha J_{x}\right)$, implying that the top keeps rotating at a constant rate; however, when *r* is odd and s=2, we have $U^{2}=1$, suggesting that the motion is periodic of period two as well.

### 3.2. The Pseudoclassical Limit of the Kicked Top

For the kicked rotor, a pseudoclassical theory has been developed to address the quantum dynamics via a classical map, the so-called pseudoclassical limit, when the system parameter is close to the resonance condition that *T* is an integer multiple of 2π (i.e., s=2) [25,26]. In the following, we attempt to extend this theory to the kicked top and study its behavior for β=4jπr/s+δ, where δ (incommensurate to π) is a weak perturbation to the resonance condition that is unnecessarily limited to the case of s=2. Suppose that the current state is |Θ,Φ〉 and its classical counterpart is $\left(\Theta ,\Phi \right)$; our task is to figure out the one-step evolution result for the latter by analogy based on the quantum evolution.

For β=4jπr/s+δ, the Floquet operator (Equation 1) can be rewritten as
(4)$$\begin{array}{cc} U & =\exp\left(-i\frac{4j\pi \frac{r}{s}+\delta }{2j}J_{z}^{2}\right)\exp\left(-i\alpha J_{x}\right)\\ \; & =\exp\left(-i2\pi \frac{r}{s}J_{z}^{2}\right)\exp\left(-i\frac{\delta }{2j}J_{z}^{2}\right)\exp\left(-i\alpha J_{x}\right). \end{array}$$

Remarkably, the last two terms are exactly the Floquet operator of the kicked top but with the kicking strength δ instead. As shown in previous studies, when δ is small, the quantum dynamics that the last two operators represent can be well mapped to the classical kicked top (with the kicking strength δ) in the semi-classical limit. As a consequence, the classical counterpart of the last two operators is to map $\left(\Theta ,\Phi \right)$ into the intermediate state
(5)$$\left({\tilde{\Theta }}_{\delta },{\tilde{\Phi }}_{\delta }\right)=F\left(\Theta ,\Phi ;\alpha ,\delta \right),$$
where the subscript δ at the l.h.s. indicates the kicking strength for the sake of clearness.

Vice versa, for the corresponding quantum evolution, due to the solid quantum–classical correspondence in the semi–classical limit, we assume that these two operators map the coherent state |Θ,Φ〉 into that of $|{\tilde{\Theta }}_{\delta },{\tilde{\Phi }}_{\delta }\rangle$. Then, the remaining problem is to find out the classical counterpart of the result when the first operator at the r.h.s. of Equation (Equation 4) applies to this intermediate state $|{\tilde{\Theta }}_{\delta },{\tilde{\Phi }}_{\delta }\rangle$. The result is (see the derivation in Appendix A)
(6)$$\exp\left(-i2\pi \frac{r}{s}J_{z}^{2}\right)|{\tilde{\Theta }}_{\delta },{\tilde{\Phi }}_{\delta }\rangle =\sum_{l=0}^{s-1}G_{l}|{\tilde{\Theta }}_{\delta },{\tilde{\Phi }}_{\delta }+\frac{2\pi r}{s}l\rangle ,$$
where $G_{l}$ is the Gaussian sum
$$G_{l}=\frac{1}{s}\sum_{k=0}^{s-1}\exp\left(-i\frac{2\pi r}{s}k\left(k-l\right)\right).$$

The physical meaning of Equation (Equation 6) is clear: the intermediate coherent state $|{\tilde{\Theta }}_{\delta },{\tilde{\Phi }}_{\delta }\rangle$ is mapped into *s* coherent states located along the line of $\Theta ={\tilde{\Theta }}_{\delta }$, each of which has an amplitude given by a Gaussian sum. Note that these *s* coherent states are not necessarily independent; some of them may correspond to the same coherent state, if their *l* values lead to the same angle of Δ=2πrl/s mod 2π. Suppose that there are N different such angles in total and denote them as $\Delta_{k}$, k=1,⋯,N; then, Equation (Equation 6) can be rewritten as
(7)$$\exp\left(-i2\pi \frac{r}{s}J_{z}^{2}\right)|{\tilde{\Theta }}_{\delta },{\tilde{\Phi }}_{\delta }\rangle =\sum_{k=1}^{N}A_{k}|{\tilde{\Theta }}_{\delta },{\tilde{\Phi }}_{\delta }+\Delta_{k}\rangle .$$

Here, for the *k*th component coherent state, its amplitude $A_{k}$ is the sum of all $G_{l}$ whose subscript *l* satisfying $\Delta_{k}=2\pi rl/s$ mod 2π.

In the semi-classical limit j→∞, a coherent state reduces to a point in the phase space. Given this, we can give Equation (Equation 7) a classical interpretation as the following: the point (Θ,Φ) is mapped into a set of N points and meanwhile each point is associated with a complex “amplitude”. These two features make the situation here distinct from the previous pseudoclassical theory for the kicked rotor, where a point is mapped only to another point and no complex amplitude is involved. Thus, formally, the pseudoclassical map that we seek can be expressed as
(8)$$M:\left(\Theta ,\Phi \right)\to \{\left[\left({\tilde{\Theta }}_{\delta },{\tilde{\Phi }}_{\delta }+\Delta_{k}\right);A_{k}\right],k=1,\cdots ,N\}.$$

This is the key result of the present work. As illustrated in the next section, it does allow us to predict the quantum dynamics in such a pseudoclassical way. Here, we emphasize that the amplitudes $\left\{A_{k}\right\}$ are crucial to this end. Specifically, $|A_{k}{|}^{2}$ has to be taken as the weight of the *k*th point to evaluate the expected value of a given observable. Moreover, the phases encoded in these amplitudes have to be considered simultaneously to correctly trace the quantum evolution.

## 4. Applications of the Pseudoclassical Theory

In this section, we check the effectiveness of the pseudoclassical map by comparing its predictions with that obtained directly with the quantum Floquet operator. In general, if a point is mapped into N>1 points at each step, then the number of points that we have to deal with would increase exponentially. Therefore, in practice, it would be prohibitively difficult to apply it for any arbitrarily given parameters. However, fortunately, for some quantum resonance parameters, N could be small, and, under certain conditions, e.g., if α is an integer multiple of π/2, coherent cancellation may suppress the increase in the number of points (see the second subsection below). In such cases, the application of the pseudoclassical theory can be greatly simplified. Here, we consider two such cases as illustrating examples, i.e., β=2jπ+δ and β=jπ+δ, respectively.

### 4.1. Case I: β=2jπ+δ

For this case, we can show (see Appendix B) that N=1, i.e., the pseudoclassical dynamics evolves the point (Θ,Φ) into another single point as
(9)$$M:\left(\Theta ,\Phi \right)\to \left({\tilde{\Theta }}_{\delta },{\tilde{\Phi }}_{\delta }+\pi \right),$$
with the corresponding amplitude $A_{1}=1$.

With Equation (Equation 9) in hand, we are ready to predict the quantum properties. The most relevant quantities could be the expected values of angular momentums. In Figure 1, their dependence on time is shown for a randomly chosen initial condition. The corresponding quantum results for three different values of *j* are plotted together for comparison. It can be seen that the pseudoclassical results agree very well with the quantum ones, and, as expected, as *j* increases, the agreement improves progressively. It shows that, indeed, the pseudoclassical limit captures the quantum motion successfully.

The agreement illustrated in Figure 1 does not depend on α. However, if α is an integer multiple of π/2, the system would have an additional interesting property. Namely, its quantum entanglement entropy would remain synchronized with that of the system that has a kicking strength of β=δ instead [18]. Since, for such an α value, the good agreement between the pseudoclassical and the quantum evolution remains equally, we can probe this interesting phenomenon from the pseudoclassical perspective. In fact, by following Equation (Equation 9) and taking into account the extra symmetry introduced by such an α value, we can show that this synchronization in the entanglement entropy roots in the synchronization of their dynamics (see the following and the next section). The latter has a period of four (two) when α is an odd (even) multiple of π/2. In Appendix B, the pseudoclassical dynamics is detailed for the representative example where α=π/2. For this case, in terms of the expected value of angular momentums, denoted as $J_{x}$, $J_{y}$, and $J_{z}$ as well, without confusion, the connection between these two systems at a given time *n* can be made explicitly as follows:(10)$$\begin{array}{c} J_{x}\left(n;\beta \right)=\left\{\begin{array}{cc} J_{x}\left(n;\delta \right), & \;\mathrm{mod}\;(n,4)=0\;\mathrm{or}\;2;\\ -J_{x}\left(n;\delta \right), & \;\mathrm{mod}\;(n,4)=1\;\mathrm{or}\;3, \end{array}\right.\\ J_{y}\left(n;\beta \right)=\left\{\begin{array}{cc} J_{y}\left(n;\delta \right), & \;\mathrm{mod}\;(n,4)=0\;\mathrm{or}\;3;\\ -J_{y}\left(n;\delta \right), & \;\mathrm{mod}\;(n,4)=1\;\mathrm{or}\;2, \end{array}\right.\\ J_{z}\left(n;\beta \right)=\left\{\begin{array}{cc} J_{z}\left(n;\delta \right), & \;\mathrm{mod}\;(n,4)=0\;\mathrm{or}\;1;\\ -J_{z}\left(n;\delta \right), & \;\mathrm{mod}\;(n,4)=2\;\mathrm{or}\;3. \end{array}\right. \end{array}$$

Note that the angular momentum values at the l.h.s. and the r.h.s. are for the system with kicking strength β=2jπ+δ and δ, respectively.

To check this prediction, we compare the numerical results of the quantum evolution of the two systems. The results are presented in Figure 2, where not only the four-step synchronization but also the details of the intermediate states can be recognized immediately. We can also make a close comparison of these two systems by visualizing their quantum evolution in the phase space with the Husimi distribution [29]. At a given point (Θ,Φ) in the phase space, the Husimi distribution P(Θ,Φ) is defined as the expectation value of the density matrix ρ with respect to the corresponding spin coherent state, i.e.,
$$P\left(\Theta ,\Phi \right)=\frac{2j+1}{4\pi }\langle \Theta ,\Phi |\rho |\Theta ,\Phi \rangle .$$

The results for β=2 and β=2jπ+2 at four different times are shown in Figure 3 and Figure 4, respectively. It can be seen that, when n=1, the centers of the two wavepackets only differ by an angle of π in Φ, while, when *n* = 2, they become symmetric with respect to (Θ,Φ)=(π/2,π). When *n* = 4 and 8, the two wavepackets are indistinguishable, which is a sign that the two systems are synchronized.

Obviously, all these numerical checks have well corroborated the effectiveness of our pseudoclassical analysis.

### 4.2. Case II: β=jπ+δ

Now, let us consider a more complex case, i.e., β=jπ+δ. For this case, N=2 and the pseudoclassical map is (see Appendix C)
(11)$$\begin{array}{c} M:\left(\Theta ,\Phi \right)\to \left\{\begin{array}{c} \left({\tilde{\Theta }}_{\delta },{\tilde{\Phi }}_{\delta }+\pi \right);\;\;A_{1},\\ \left({\tilde{\Theta }}_{\delta },{\tilde{\Phi }}_{\delta }\right);\;\;A_{2}, \end{array}\right. \end{array}$$
with $A_{1}=\left(1+i\right)/2$ and $A_{2}=\left(1-i\right)/2$. It implies that, after each step, a point will be mapped into two points at the same probability but with different phases. This map looks simple, but as N=2, if we use it to predict the quantum evolution, the points will proliferate in time so that, in practice, we can trace the quantum motion for a few steps only. Interestingly, this fact might explain why the quantum motion would be complicated from a new perspective.

Nevertheless, for some special values of α, due to the coherent effect, the newly generated points after a step of iteration may overlap and cancel each other out, making the number of points increase more slowly. An intriguing example is that discussed in the previous subsection, i.e., where α is an integer multiple of π/2. Again, for such an α value, the system is brought to synchronization with the system of kicking strength β=δ as well, but with instead a period of eight (four) if α is an odd (even) multiple of π/2. To be explicit, for α=π/2, the connections between the two systems are presented in Appendix C. In terms of the expected value of angular momentums, we have
(12)$$\begin{array}{cc} J_{x}\left(n;\beta \right) & =\left\{\begin{array}{cc} J_{x}\left(n;\delta \right), & \;\;\mathrm{mod}\;(n,8)=0\;\mathrm{or}\;4;\\ 0, & \;\;\mathrm{else}, \end{array}\right.\\ J_{y}\left(n;\beta \right) & =\left\{\begin{array}{cc} J_{y}\left(n;\delta \right), & \;\mathrm{mod}\;(n,8)=0\;\mathrm{or}\;7;\\ -J_{y}\left(n;\delta \right), & \;\mathrm{mod}\;(n,8)=3\;\mathrm{or}\;4;\\ 0, & \;\mathrm{else}, \end{array}\right.\\ J_{z}\left(n;\beta \right) & =\left\{\begin{array}{cc} J_{z}\left(n;\delta \right), & \;\mathrm{mod}\;(n,8)=0\;\mathrm{or}\;1;\\ -J_{z}\left(n;\delta \right), & \;\mathrm{mod}\;(n,8)=4\;\mathrm{or}\;5;\\ 0, & \;\mathrm{else}. \end{array}\right. \end{array}$$

The simulation results of the quantum angular momentums for β=jπ+δ are shown in Figure 2 as well; they support this derivation convincingly.

Based on the pseudoclassical dynamics, we find that the initial point will be mapped into two and then four points after the first and the second iteration, respectively. However, after the third iteration, the points do not become eight as expected; rather, these eight points can be divided into four pairs and the two points in each pair overlap with each other. Moreover, two of these four points disappear in effect as the resultant total amplitude for each of them turns out to be zero (see Appendix C). Thus, only two points remain, and after the fourth iteration, these two points further merge into one. As a consequence, the number of points varies in time with a period of four. The results for the Husimi distribution given in Figure 5 are in good agreement with this analysis. Comparing this with the results for β=δ in Figure 3, we can see that after the fourth iteration, there is only one wavepacket of the same shape in both cases, but their positions are different. Only after the eighth iteration, the two wavepackets are identical, which explains why the synchronization period should be eight.

## 5. The Dynamical Entanglement

The dynamical entanglement of the quantum kicked top has been studied carefully in recent years. We discuss this issue in this section by taking advantage of the pseudoclassical results obtained in the previous section.

For the quantum kicked top, its momentum can be represented by 2j qubits, or a collection of 2j spin-1/2 identical particles. If the initial state of the system is symmetric under permutations for identical qubits, this permutation symmetry will be preserved, as it is respected by the action of the Floquet operator of the kicked top. As a consequence, the expected spin value for any single qubit of these 2j identical qubits is $s_{\gamma }=J_{\gamma }/\left(2j\right)$, where $J_{\gamma }$ is the expected momentum value of the top and γ=x,y, and *z* [30].

On the other hand, though various bipartite entanglement measures have been suggested, it has been shown that they are qualitatively equivalent. Thus, we adopt the measure considered the most frequently, i.e., the bipartite entanglement between any qubit and the subsystem made up of the remaining 2j−1 qubits. This entanglement is usually quantified by computing the linear entropy $S=1-\mathrm{Tr}(\rho_{1}^{2})$, where $\rho_{1}$ denotes the reduced density operator for a single qubit. As $\rho_{1}$ is a 2×2 operator, it can be expressed as $\rho_{1}=1/2+\sum_{\gamma }s_{\gamma }\sigma_{\gamma }$, given the expected spin value $s_{\gamma }$. Here, $\sigma_{\gamma }$ is the Pauli operator. By substituting $s_{\gamma }=J_{\gamma }/\left(2j\right)$, we have $\rho_{1}=1/2+\sum_{\gamma }J_{\gamma }\sigma_{\gamma }/\left(2j\right)$ in terms of $J_{\gamma }$ instead [30]. It follows that
(13)$$S=\frac{1}{2}\left(1-\frac{{\left[J_{x}\right]}^{2}+{\left[J_{y}\right]}^{2}+{\left[J_{z}\right]}^{2}}{j^{2}}\right),$$
which has a well-defined classical counterpart and is easy to compute numerically. Here, ${\left[J_{\gamma }\right]}^{2}$ represents the square of the expected value of angular momentum $J_{\gamma }$.

For the two cases close to the quantum resonance condition discussed in the previous section, we can see immediately how their linear entropy is related to that of the case β=δ based on the pseudoclassical analysis. First, for β=2jπ+δ, from Equation (Equation 10), we have that ${\left[J_{\gamma }\left(n;\beta \right)\right]}^{2}={\left[J_{\gamma }\left(n;\delta \right)\right]}^{2}$ at any time *n*; hence, S(n) must coincide with that for the system of β=δ throughout. However, for β=jπ+δ, from Equation (Equation 12), we know that $\sum_{\gamma }{\left[J_{\gamma }\left(n;\beta \right)\right]}^{2}=\sum_{\gamma }{\left[J_{\gamma }\left(n;\delta \right)\right]}^{2}$ only when mod(n,8)=0 or 4. Therefore, we may expect a synchronization of period four in the linear entropy. In addition, from Equation (Equation 12), we also know that $\sum_{\gamma }{\left[J_{\gamma }\left(n;\beta \right)\right]}^{2}=0$ when mod(n,8)=2 or 6, suggesting that the linear entropy should reach its maximal value repeatedly in a period of four as well. As for the case β=δ itself, because the quantum motion can be approximated by the semi-classical limit if δ is small, the time dependence of the linear entropy can be predicted based on the classical dynamics [15]. In a chaotic region of the phase space, S(n) should increase linearly before saturation; otherwise, it proceeds in a logarithmic law that features the regular motion. The numerical results of the linear entropy for these three cases and two representative initial states are presented in Figure 6, which meet all these expectations.

Another interesting and related quantity that has been intensively investigated is the time-averaged entanglement entropy. If the semi-classical limit exists, it is used to estimate the equilibrium value that the corresponding classical system tends to. For the linear entropy, it is defined as
(14)$$S_{\tau }=\frac{1}{\tau }\sum_{n=1}^{\tau }S\left(n\right),$$
where τ is a sufficiently long time. Interestingly, it was found that the contour plot of $S_{\tau }$ can well capture the characteristics of the phase space portrait of the corresponding classical system [19,20]. The reason is that, given the semi-classical limit, for all initial coherent states centered on the same classical trajectory, by definition, $S_{\tau }$ should be the same in the long average time limit. As such, $S_{\tau }$ can be used to distinguish the trajectories. As an illustration, in Figure 7a, the contour plot of $S_{\tau }$ for the case β=2, where the semi-classical limit holds well, is shown. The corresponding phase space portrait of its classical counterpart is shown in Figure 8. The similarity between them is easy to recognize.

For the two cases close to the quantum resonance condition, the semi-classical limit breaks. However, as the pseudoclassical limit exists, it allows us to use $S_{\tau }$ to probe the corresponding pseudoclassical systems. In Figure 7b, the contour plot for β=2jπ+2 is shown. It is identical to that for β=2, because the two cases share the same linear entropy at every step. However, it is worth noting that, although, from Equation (Equation 9), we know that any trajectory of the pseudoclassical system for β=2jπ+δ is related to one of the semi-classical system for β=δ, they are not located at the same positions in their respective phase spaces (see Figure 3 and Figure 4, for example). This suggests that $S_{\tau }$ is still equally helpful for obtaining an overall sense of the pseudoclassical dynamics, but some details could be missed inevitably by definition.

More interesting is the case for β=jπ+δ. For the pseudoclassical dynamics, the concept of the conventional trajectory does not apply any longer, because, at a given time, the number of points in the phase space can be multiple, and they have their respective complex amplitudes. Regardless of this fact, as Figure 7c shows, $S_{\tau }$ works well again for schematizing the pseudoclassical dynamics. For example, due to the periodic synchronization with the case of β=δ, we may expect that the chaotic and regular regions of the former are exactly those of the latter, respectively. This is indeed the case, which can be seen by comparing Figure 7a,c. Alternatively, if we compute $S_{\tau }$ by taking the average of S(n) once every four steps (the period of synchronization), the results should be the same exactly for both cases.

## 6. Summary and Discussion

In summary, we have introduced the quantum resonance condition into the kicked top model. In order to study the behavior of the quantum kicked top detuned from the quantum resonance condition, we have established the corresponding pseudoclassical theory. By analytical and numerical studies, we have shown that this theory is effective. In particular, when being applied to discuss the dynamical entanglement, the properties of the quantum kicked top are successfully predicted based on the pseudoclassical dynamics. Our results also suggested that the time-averaged entanglement entropy is still a powerful tool for grasping the pseudoclassical dynamics.

The suggested pseudoclassical scheme is distinct from the one originally introduced in the kicked rotor that works only near the special quantum resonance condition that *T* is an integer multiple of 2π. To make this explicit, let us extend our scheme to the kicked rotor and compare it with the original pseudoclassical limit. The result is similar to Equation (Equation 8):(15)$$M:\left(p,\theta \right)\to \{\left[\left({\tilde{p}}_{\delta },{\tilde{\theta }}_{\delta }+\Delta_{k}\right);A_{k}\right],k=1,\cdots ,N\},$$
but, here, (p,θ) is the classical counterpart and the center of the coherent state |p,θ〉 for the kicked rotor, and $\left({\tilde{p}}_{\delta },{\tilde{\theta }}_{\delta }\right)$ is that (p,θ) is mapped to by the classical kicked rotor dynamics with the kicking strength δK. For T=2kπ+δ (*k* is an integer), it gives exactly the original pseudoclassical result. However, this scheme also applies when the system is close to other quantum resonances, making it more general than the original one.

In practice, the main challenge for applying this scheme is the same as that encountered in the kicked top, i.e., the rapid proliferation of phase space points. However, if we introduce an additional symmetry into the system, i.e., the translation invariance for θ→θ+2π/w, where w=s/2 for an even *s* and w=s for an odd *s*, respectively, it is found that the coherent cancellation mechanism works efficiently so that the proliferation can be greatly suppressed. The translation invariance can be fulfilled by replacing the potential cosθ in the Floquet operator $U_{R}$ with cos(wθ). Such a favorable property makes the kicked rotor even more advantageous than the kicked top for demonstrating the pseudoclassical dynamics. A detailed discussion will be published elsewhere [31].

With the pseudoclassical theory in hand, some interesting problems could be investigated further. For example, it may be applied to study the entanglement in the kicked top with weak measurements of one or several qubits [32] to identify the measurement effect from a different perspective. Moreover, the proliferation of the phase space points is in clear contrast with the conventional classical dynamics. As it is inherited from the quantum evolution, logN might be taken as a complexity measure of the quantum dynamics. For some previously studied problems in the double kicked rotor and top, such as Hofstadter’s butterfly spectrum [33,34,35] and exponential and superballistic wavepacket spreading [36,37], it may help us to gain a deeper understanding. For some frontier topics mentioned in the Introduction, this theory may find applications as well.

## Figures and Tables

**Figure 1 entropy-24-01092-f001:**
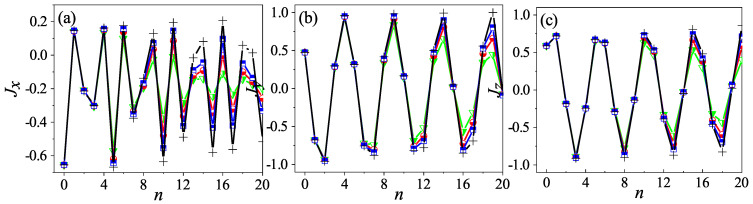
The time dependence of the expected value of angular momentum $J_{x}$ (**a**), $J_{y}$ (**b**), and $J_{z}$ (**c**), respectively, for α=1 and β=2jπ+2. The black crosses are for the results by the pseudoclassical map (Equation (Equation 9)). The green triangles, the red circles, and the blue squares are for the quantum results with j=100, 200, and 400, respectively. For the initial state (Θ,Φ) and |Θ,Φ〉, Θ=0.8π and Φ=0.3π.

**Figure 2 entropy-24-01092-f002:**
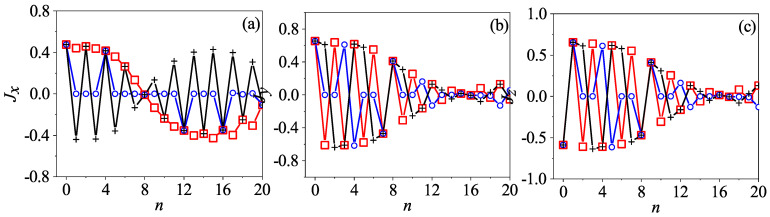
The time dependence of the expected value of angular momentum $J_{x}$ (**a**), $J_{y}$ (**b**), and $J_{z}$ (**c**), respectively. The same as Figure 1 but for the quantum kicked top with kicking strength β=2jπ+2 (red squares), β=jπ+2 (blue circles), and β=2 (black crosses), respectively. Here, α=π/2, j=400, and the initial state is |Θ,Φ〉=|0.7π,0.3π〉.

**Figure 3 entropy-24-01092-f003:**
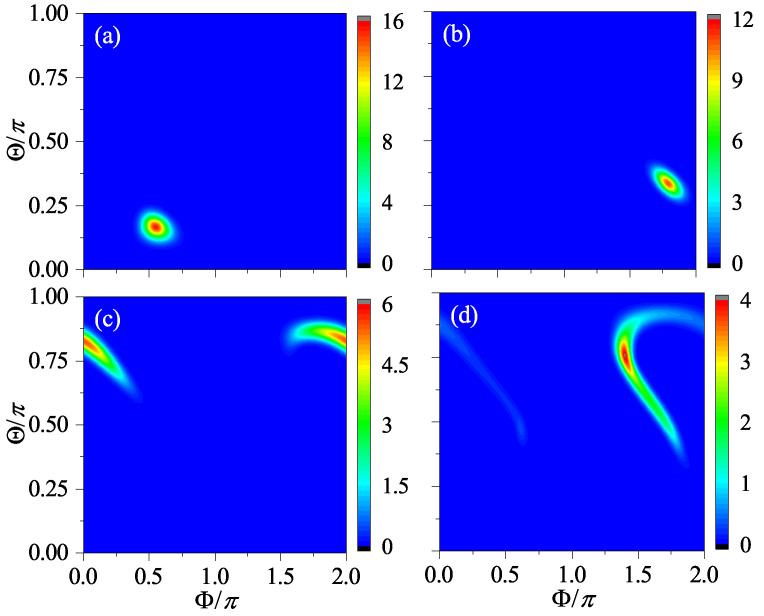
The Husimi distribution for β=2 at the time n=1 (**a**), n=2 (**b**), n=4 (**c**), and n=8 (**d**), respectively. Here, α=π/2, j=100, and the initial state is |Θ,Φ〉=|π/2,π/3〉.

**Figure 4 entropy-24-01092-f004:**
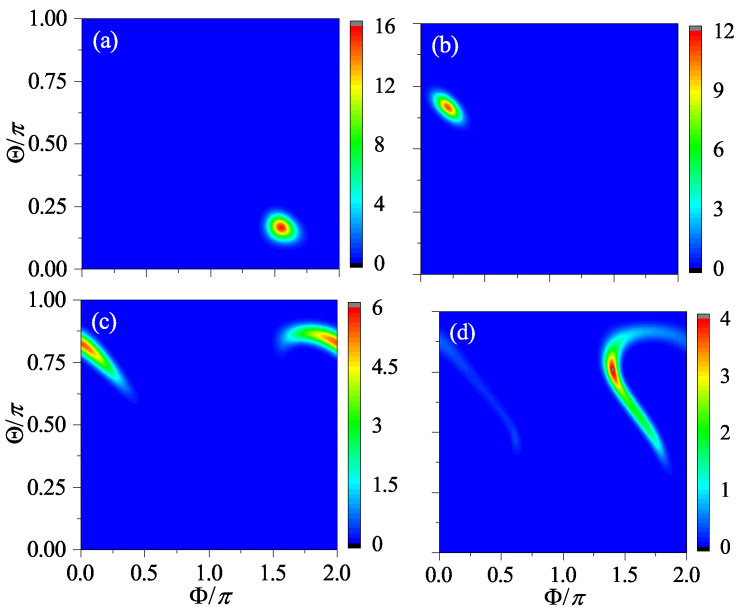
The Husimi distribution for β=2 at the time n=1 (**a**), n=2 (**b**), n=4 (**c**), and n=8 (**d**), respectively. The same as Figure 3 but for β=2jπ+2 instead (other parameters remain unchanged).

**Figure 5 entropy-24-01092-f005:**
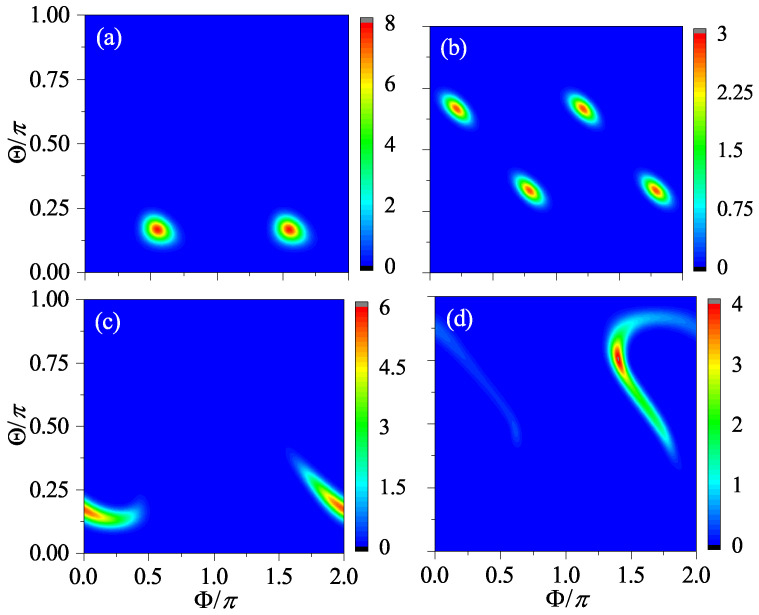
The Husimi distribution for β=2 at the time n=1 (**a**), n=2 (**b**), n=4 (**c**), and n=8 (**d**), respectively. The same as Figure 3 and Figure 4 but for β=jπ+2. All other parameters are the same as in the former two figures.

**Figure 6 entropy-24-01092-f006:**
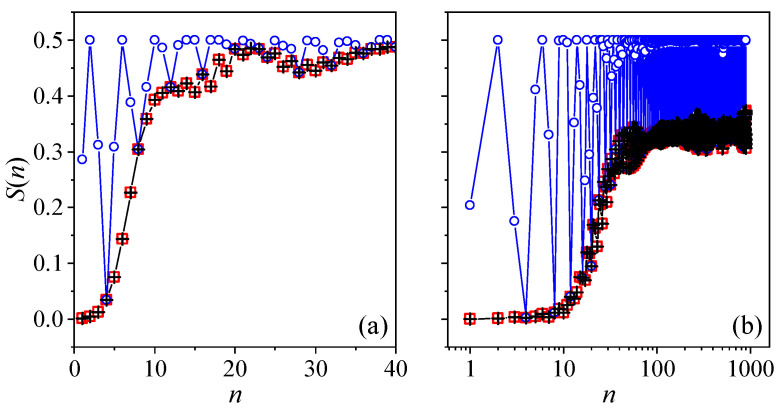
The linear entropy as a function of time for the initial condition |Θ,Φ〉=|0.7π,0.3π〉 (**a**) and |0.7π,0.6π〉 (**b**), respectively. The classical counterparts of these two states are in the chaotic and regular region of the phase space, respectively, for the classical kicked top of β=2. In both panels, the red squares, the blue circles, and the black crosses are for, respectively, β=2jπ+2, β=jπ+2, and β=2. For all the cases, α=π/2 and j=400.

**Figure 7 entropy-24-01092-f007:**
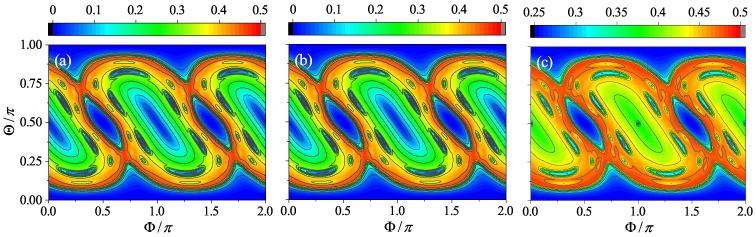
The contour plot of the time-averaged entanglement entropy, $S_{\tau }$ for kicking strength β=2 (**a**), β=2jπ+2 (**b**), and β=jπ+2 (**c**), respectively. Here, α=π/2, j=400, τ=300, and a grid of 201×201 initial coherent states is simulated for each case.

**Figure 8 entropy-24-01092-f008:**
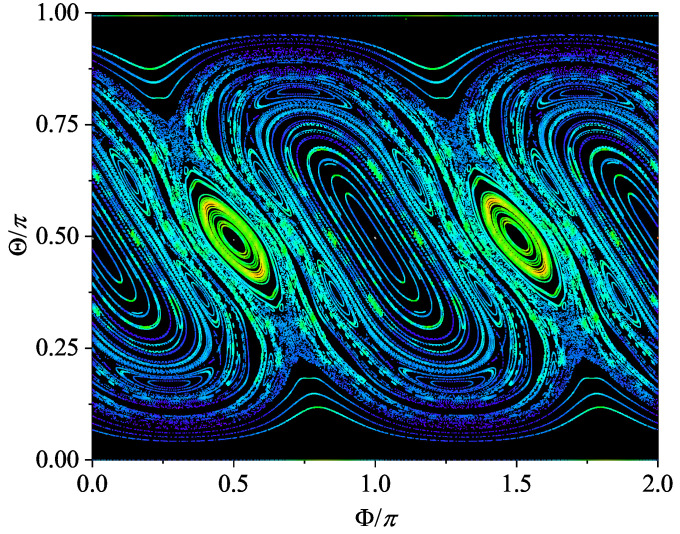
The phase space portrait of the classical kicked top with α=π/2 and β=2.

## Data Availability

Not applicable.

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
