# Peer review of "Pseudoclassical Dynamics of the Kicked Top"

_entropy, 2022, doi:10.3390/e24081092_

Round 1
Reviewer 1 Report
This work extends the previous concepts of quantum resonance and pseudoclassical limit, which are established in the study of kicked rotor, to the kicked top model and shows that the dynamical behaviors of the quantum kicked top perturbed from the quantum resonance condition for some particular parameters can be well captured by the pesudoclassical theory. By applying the pesudoclassical theory to discuss the entanglement entropy evolution in the kicked top model, the authors demonstrate a good agreement between the numerical results and the predictions of the pesudoclassical theory.
The paper is very well written and explains each step in the analysis carefully. I thus suggest publication of the manuscript as it stands.
Typos: In the first line of Eq. (4), on the right hand side of equation, the first term should have $J_z^2$.

Reviewer 2 Report
This paper studies the dynamics of the kicked top near a resonance condition that allows for an efficient asymptotic description. In contrast to analogous descriptions in other models, this results in a splitting of trajectories, which leads to interesting new effects. This is elaborated by looking at special cases, both directly for the evolution as well as through the entanglement after reinterpretation as a many-body system.
This is a very nicely written paper with an conceptually interesting main result and direct applications to the model. I am very happy to recommend its publication.
Minor comments:
Jz^2 seems to be missing in the first exponential on the first line of Eq (4).
Are there any more concrete signatures of the mixed phase space dynamics at moderate delta?
Describe briefly how the reduced density matrix of 1 spin is obtained from the 2j+1 component vector.
A physical situations this could be directly applied to is the entanglement in the kicked top with weak measurements of one or several qubit, see J. Phys. A 55, 214001 (2022).
The authors should check Ref 13 (this may be a mix up of authors from one paper and a reference from another paper, https://link.springer.com/article/10.1007/JHEP02(2016)004 ).
